Living (stained) foraminifera in the Lesser Syrtis (Tunisia): influence of pollution and substratum

El Kateb Akram 1
Beccari Valentina 1
Stainbank Stephanie 1
Spezzaferri Silvia silvia.spezzaferri@unifr.ch 1
Coletti Giovanni 2
1 Department of Geosciences, University of Fribourg , Fribourg , Switzerland
2 Department of Earth and Environmental Sciences, University of Milan—Bicocca , Milano , Italy
Gavrilescu Maria
Electronic publication date: 2020 Apr 6
Publication date: 2020
Volume: 8
Electronic Location ID: e8839
Received 2019 Sep 26; Accepted 2020 Mar 2
Copyright: ©2020 El Kateb et al.
Copyright year: 2020
Copyright holder: El Kateb et al.
License: This is an open access article distributed under the terms of the Creative Commons Attribution License, which permits unrestricted use, distribution, reproduction and adaptation in any medium and for any purpose provided that it is properly attributed. For attribution, the original author(s), title, publication source (PeerJ) and either DOI or URL of the article must be cited.
License URL: https://creativecommons.org/licenses/by/4.0/

Keywords: Foraminifera, Bioindicators, Taxonomy, Pollution, Tunisia

Funding: Swiss National Science Foundation 200021_149589 This work was funded by the Swiss National Science Foundation (grant 200021_149589). The funders had no role in study design, data collection and analysis, decision to publish, or preparation of the manuscript.

==============================
Foraminifera are protozoans with biomineralized tests that can be successfully used as a low cost monitoring tool to assess the health status of marine environments. Living benthic foraminiferal assemblages can provide essential information on natural and/or anthropogenic stresses and provide baseline conditions for studies on fossil material. Several studies have highlighted the negative impact of phosphate treatment industries along the Gulf of Gabes (Lesser Syrtis, Tunisia) on the marine environment. However, only a few studies, based on living (stained) benthic foraminifera, are presently available to assess environmental and/or ecological conditions in this Gulf. Thirty-eight surface sediment samples were quantitatively investigated to identify the dominant living benthic foraminiferal species and potential pollution-sensitive and stress-tolerant species. One-hundred and sixty-one species were identified, and grouped into seven clusters representing different environments within the Gulf. These groups represent polluted settings (Cluster A and B), polluted environments characterized by physicochemical variability (Cluster C), seagrass meadows and “pristine” sites (Cluster D and E) and the region subjected to major industrial impact (Cluster F). The final outlier Cluster, identified the foraminifera barren and all shallow coastal stations. A SIMPER analysis helped identify species with clear and fast responses to environmental perturbations (Ammonia tepida, Amphistegina lessonii, Brizalina striatula, Bulimina marginata, Buliminella elegantissima, Eggereloides scaber, Peneroplis perutusus, Rosalina macropora, Rosalina villardeboana, Trochammina inflata). A comparison with the measured geochemical parameters (TOC, phosphorus in the sediments and heavy metal concentrations in the seawater) has shown that the benthic foraminiferal assemblages are mainly linked to phosphorus, TOC, As and Cd pollution. We also provide here the first compilation of the identified living species in the Lesser Syrtis, their synonyms and digital images of important species.

Introduction

The phosphate treatment industry started in Tunisia (Lesser Syrtis) in the second half of the 20th century. The high degree of pollution generated by this industry, within the Gulf of Gabes, is well documented. As a result of this phosphate pollution, the marine environment within the Gulf of Gabes has been deeply affected since the beginning of the industrial exploitations. The first production unit was created at Sfax in 1952, followed by the industrial complex of Gabes in 1972, while the last industrial complex was created at Skhira in 1988. These industries generate a large amount of waste products e.g., phosphogypsum (PG). For each ton of phosphoric acid that is produced, five tons of PG are generated (Zairi & Rouis, 1999; Tayibi et al., 2009). This waste product is well known to contain several types of pollutants such as heavy metals, fluorine, phosphorus and even radioactive isotopes (Rutherford, Dudas & Samek, 1994; Zairi & Rouis, 1999; Pérez-López, Alvarez-Valero & Nieto, 2007; Ajam et al., 2009; El Afifi et al., 2009; Tayibi et al., 2009; Ajmal et al., 2014). Phosphogypsum is generally stored on land forming giant stacks (up to 60 m high) but at Gabes, all the industrial waste (including PG, industrial sludge, and wastewater) is directly discharged into the sea.

This industrial pollution has heavily contaminated the marine sediments (El Zrelli et al., 2015; Gargouri et al., 2011; Wali et al., 2013; Ayadi, Aloulou & Bouzid, 2015; Mkawar et al., 2007) and seawater (Darmoul, Hadj-Ali & Vitiello, 1980; El Zrelli et al., 2018). Resultant heavy metals bioaccumulation in marine fauna is documented (Rabaoui et al., 2014; Messaoudi et al., 2009). The decline of the coral Cladocora caespitosa in the region (El Kateb et al., 2016) and of macrobenthic faunas (Zaouali, 1993; Pérez Domingo, Castellanos & Junoy, 2008; El Lakhrach et al., 2012) can be also attributed to the pollution.

Aside from impacting the phosphorus and heavy metal contents in the sediments and water, additional forms of pollution are generated by the phosphate industry including increased siltation of the seafloor. At the beginning of the 20th century, a large part of the Gulf of Gabes was colonized by the seagrass Posidonia oceanica (El Zrelli et al., 2017 and references therein). Ben Brahim et al. (2010) estimated a 90% loss of this seagrass cover since 1960, which has consequently caused prominent siltation. A large area of the Gulf is today covered by silty and muddy sediments (El Kateb et al., 2018b).

During the last decades benthic foraminifera have been widely used to investigate environmental conditions (e.g., Resig, 1960; Watkins, 1961), and their application has been recently extended to assess the ecological status (e.g., Murray, 2006; Martínez-Colón, Hallock & Green-Ruíz, 2009; Hallock et al., 2003; Hallock, 2012; Schönfeld et al., 2012; Dimiza et al., 2016; Martínez-Colón et al., 2018, and references therein). Compared to other macro-and micro-organisms, benthic foraminifera are advantageous because (i) they occur in marine environments all over the world; (ii) a small volume of sediment sample is needed to use them in environmental assessments due to the high foraminiferal densities (up to thousand specimens per 100 cm3 of sediment) and (iii) their short life cycle allows them to rapidly react to external stressors, e.g., pollutant contamination, anomalously high organic matter supply (eutrophication/nutrification) (e.g., Schönfeld et al., 2012; Alve et al., 2016; Jorissen et al., 2018, and reference therein) and/or thermal stress. Their added value is also the production of a mineralized test that is preserved in the sedimentary archives, which provides the possibility to reconstruct paleoenvironmental and paleoclimatic changes.

Numerous studies, based on living foraminifera, are used to highlight faunal changes as a response to short-term stresses (e.g., Morvan et al., 2004; Denoyelle et al., 2010). These investigations require the living cell to be stained at the time of sampling. The FOBIMO Initiative was instigated to meet the requirements of the Marine Strategy Framework Directive-MSFD (European Parliament, 2008). It provided, for the first time, a standardized protocol for sampling and treating sediments for biomonitoring studies based on living (stained) benthic foraminifera (Schönfeld et al., 2012). This protocol is presently accepted and applied to assess the impacts of pollution on marine environments (e.g., Buosi et al., 2013; Titelboim et al., 2016) it has therefore, been applied in this study.

Only a few studies, based on benthic foraminifera, have been conducted in the Gulf of Gabes. The investigation of Aloulou, EllEuch & Kallel (2012) along the northern coast of the Gulf and of Ayadi et al. (2016) in the proximity of the industries at Gabes are based on total benthic foraminiferal assemblages. As these two studies use total assemblages they cannot be considered as representative of the current ecological and environmental conditions. An additional study on living benthic foraminifera in Tunisia from Martins et al. (2015) and Martins et al. (2016a) concerns the Bizerta lagoon, a site with a very limited access to the open Mediterranean, located in northern Tunisia, far from the influence of the Gabes industries.

The aim of the present research is to document variations in benthic foraminiferal assemblage composition in relation to the pollution sources, to assess the response of living (stained) benthic foraminifera to the pollution produced by the phosphate industries in the Gulf of Gabes and to provide a comparison with a more pristine site (Djerba Island).

Materials and Methods

Study area

The investigated area is located in the Gulf of Gabes, a 90 km wide and 100 km long embayment in the Mediterranean Sea. It is delimited by the Kerkhenna Islands to the north and Djerba Island to the south. Its bathymetry is gently sloping from the coast to around 150 km with a ±50 m water depth. Within this region, tides are the highest of the entire Mediterranean Sea reaching 1.7 m (Aloulou, EllEuch & Kallel, 2012). The thirty-eight investigated stations can be subdivided into three groups (Fig. 1, Table 1): Gabes (16 stations, from GBS-01 to GBS-16) and Djerba transects (15 stations, from DJB-01 to DJB-15); Coastal stations (7 stations, from CST-01 to CST-07).

Figure 1 Location map.

Location map of the Gulf of Gabes and position of the investigated sites, showing the Gabes and Djerba transects and coastal stations (CST). Modified after El Kateb et al. (2018c).

Table 1 Coordinates.

Geographic coordinates and water depths of the sampled stations.

Location	Water depth (m)	Coordinates (latitude and longitude)	
Gabes Transect			
GSB-01	4.5	N33°54′36.66″/E10°6′34.74″	
GSB-02	7.3	N33°54′52.32″/E10°7′11.76″	
GSB-03	9.6	N33°55′9.54″/E10°7′44.10″	
GSB-04	9.1	N33°55′29.82″/E10°8′15.36″	
GSB-05	12	N33°55′48.00″/E10°8′46.92″	
GSB-06	12.9	N33°56′7.74″/E10°9′15.72″	
GSB-07	14.4	N33°56′30.60″/E10°9′46.74″	
GSB-08	15	N33°56′57.24″/E10°10′18.84″	
GSB-09	18	N33°57′16.20″/E10°10′53.82″	
GSB-10	19.5	N33°57′29.34″/E10°11′36.24″	
GSB-11	19.5	N33°57′40.08″/E10°12′9.30″	
GSB-12	15,4	N33°57′56.64″/E10°12′47.04″	
GSB-13	19.5	N33°58′9.66″/E10°13′22.20″	
GSB-14	17.6	N33°58′14.04″/E10°13′58.86″	
GSB-15	19.5	N33°58′27.18″/E10°14′39.18″	
GSB-16	18	N33°58′53.16″/E10°16′15.60″	
Djerba Transect			
DJB-01	5.1	N33°52′13.26″/E10°58′22.02″	
DJB-02	6.4	N33°52′33.12″/E10°59′1.86″	
DJB-03	8.7	N33°52′49.38″/E10°59′24.60″	
DJB-04	10.7	N33°53′3.84″/E10°59′57.84″	
DJB-05	12.7	N33°53′32.82″/E11°0′15.30″	
DJB-06	12.2	N33°53′53.34″/E11°0′31.20″	
DJB-07	14	N33°54′9.78″/E11°0′58.38″	
DJB-08	17.2	N33°54′45.54″/E11°1′21.84″	
DJB-09	17.6	N33°55′9.42″/E11°1′48.78″	
DJB-10	21.3	N33°55′21.90″/E11°2′24.12″	
DJB-11	22	N33°55′48.36″/E11°3′4.68″	
DJB-12	24	N33°56′24.12″/E11°2′59.76″	
DJB-13	25,3	N33°56′53.16″/E11°3′36.36″	
DJB-14	26	N33°57′5.64″/E11°4′1.80″	
DJB-15	26.8	N33°57′30.36″/E11°4′9.84″	
Coastal Stations			
CTS-01	<1 m	N 35°53′49.15″/E10°35′45.79″	
CTS-02	<1 m	N34°17′19.98″/E10°5′45.00″	
CTS-03	<1 m	N34°2′15.84″/E10°2′9.36″	
CTS-04	<1 m	N33°53′5.76″/E10°7′13.92″	
CTS-05	<1 m	N33°41′57.90″/E10°21′34.02″	
CTS-06	<1 m	N33°43′39.00″/E10°44′22.50″	
CTS-07	<1 m	N33°51′34.62″/E10°44′41.40″	

Sampling and samples treatment

The Djerba and Gabes transects were sampled in July 2014, perpendicular to the coastline at water depths ranging from 4.5 to 19.5 m and 5.1 to 26.8 m, respectively. The Gabes transect (GBS-01 to GBS-16) is 17.3 km long and is located between the industrial and fishing harbours of Gabes. Stations are positioned at approximately 1 km intervals. The Djerba transect (DJB-01 to DJB-15) is 13.8 km long and is located off the eastern coast of Djerba Island (El Kateb et al., 2018b). Similarly, stations are positioned approximately 1 km apart. Five sedimentary facies were identified along the Gabest transect (facies G1-G5) and three along the Djerba trasect (facies D1-D3, Fig. 2, Table 2).

Coastal stations (CST-01 to CST-07) were collected in January 2014, along the shoreline at shallow depths (<1 m) and cover 200 km of the eastern Tunisian coastline. One station is located in the Gulf of Hammamet next to El Kantaoui harbour (CST-01), four stations are located along the Gulf of Gabes from Skhira to Zarat (CST-02 to CST-05) and two stations are located along the western coast of Djerba Island (CST-06 and CST-07) (El Kateb et al., 2018b).

Sediment samples from both the Gabes and Djerba transects were collected using an Ekman-Birge box corer (15 × 15 × 30 cm), which was deployed from a small fishing boat. Sediments from coastal stations were collected by hand. All samples were collected and treated following the FOBIMO protocol (Schönfeld et al., 2012). The first centimeter of surface sediment (an area of 50 cm2) was collected in plastic bottles and placed in a rose Bengal solution (2 g/L in alcohol at 90%) for several weeks. The living (stained) foraminifera were investigated with a Nikon SMZ18 microscope, picked and placed in plummer cells, identified at species level and counted. Taxonomic identifications at species level follows Cimerman & Langer (1991), Hottinger, Halicz & Reiss (1993), Loeblich Jr & Tappan (1994), Milker & Schmiedl (2012) (Supplemental Information 1).

Figure 2 Profiles of the Gabes (A) and Djerba (B) transects showing the position of the collected samples, the sedimentary facies and the associated SIMPROF Clusters (squares), modified after El Kateb et al. (2018c).

Table 2 Facies.

Summary of the identified sedimentary facies along the Gabes and Djerba Transects (modified after El Kateb et al., 2018b).

Stations	Facies	Sediments	
GBS-01 to -02	G1	Dominant siliciclastic grains (e.g., quartz).	
GBS-03	G2	Centimetric concretions of biogenic fragments partially dissolved (e.g., bivalve, bryozoan, foraminifera and coral).	
GBS-04 to -06	G3	Carbonate sand and biogenic fragments (bryozoans, bivalves, gastropods and coral)	
GBS-07 to -08, GBS-10 to -16	G4	Fine sediment (clay) with centimetric biogenic fragments	
GBS-09	G5	Very fine sediments (clay and silt) with rare biogenic fragments.	
DJB-01 to -05	D1	Fine siliciclastic grains, Posidonia oceanica meadow	
DJB-06 to -08.	D2	Mix of sand and millimetric rounded biogenic poorly preserved fragments	
DJB-11 to -15	D3	Well-preserved biogenic fragments of bivalves, calcareous algae, gastropods and bryozoans.	

Geochemical analyses

The phosphorus sequential extraction (SEDEX) method of Ruttenberg et al. (2009) was applied to all samples. The SEDEX extraction allows an accurate quantification of the different sedimentary phosphorus reservoirs in both modern sediments and sedimentary rocks (Ruttenberg, 1992; Coletti et al., 2017). The Total Organic Carbon (TOC in wt %) was measured with Rock-Eval6 (Behar, Beaumont & Penteado, 2001). Total carbon, hydrogen and nitrogen (C, H, N) content (in wt %) were measured in all surface sediment samples using a Thermo Finnigan Flash EA 1112 gas chromatography analyser. Seawater samples were collected at the seafloor; elemental analyses (e.g., As, Cd, Cu, Ni, Fe, Cr, Li, Pb, Zn, P) were carried out by inductively coupled plasma optical emission spectroscopy (ICP-OES). More detailed descriptions of all the geochemical methods can be found in El Kateb et al. (2018c); El Kateb et al. (2018b); El Kateb et al. (2020), raw data are here presented in Supplemental Information 2.

Statistical treatment

Quantitative foraminiferal counts were treated with the Primer 7 software (Clarke et al., 2014). The data set was fourth root transformed to limit the contribution of the most abundant species (Field, Clarke & Warwick, 1982) and the Bray-Curtis (dis-)similarity calculated. A Similarity Profile (SIMPROF) cluster analysis was performed to objectively define the groups within the dendrogram (Fig. 3) and Multidimensional Scaling (MDS) plots (Fig. 4). Based on the SIMPROF grouping, a Similarity Percentage (SIMPER) analysis was run to identify discriminating taxa both within and between the groups. To assess the relationship between the biotic (foraminiferal) assemblage data and measured environmental (sediment and water) parameters a BIOENV and global BEST test (statistical significance) were performed. Following this, and using the identified best combination of environmental parameters, a LINKTREE analysis (Fig. 5) was performed to link the foraminiferal assemblage patterns with the suite of environmental parameters.

Figure 3 SIMPROF dendrogram and associated Clusters, based on the Bray–Curtis similarity matrix of fourth-root transformed living (stained) benthic foraminifera abundance data from the Lesser Syrtis.

Gray square indicates the outlier Cluster composed of the foraminifera barren and all shallow coastal stations, note that these are not considered in further statistical analyses.

Results

Six thousand, eight hundred and eighty seven living (stained) benthic foraminiferal specimens, belonging to 161 species, were identified (Supplemental Information S2 and  S3).

Figure 4 nMDS plots showing SIMPROF clusters and abundances of selected benthic foraminferal species.

Note that the nMDS plot has no dimensions and no axes and can be arbitrarily scaled, located, rotated or inverted. It gives simply the relationship of samples relative to each other (Clarke et al., 2014).

Samples collected close to the industry in Gabes (GBS-01 to 02) are dominated by Ammonia tepida and Ammonia parkinsoniana with rare specimens of Brizalina striatula, the number of species at these stations varies from 5 to 8 and the correspondent Shannon Index is around 1.5. Sample GBS-03, collected in the industrial sludge, is barren of benthic foraminifera. The other samples collected along the Gabes transect (GBS-04 to -15) contain a variable number of species from 6 (in GBS-09) to 37 (in GBS-10 and -11), with a Shannon Index of 0.8 and 2.6, respectively. In these samples the assemblages mostly consist of A. tepida, Asterigerinata mamilla, B. striatula, Bulimina alazanensis, Bulimina elegantisima, Eggerelloides scaber.

Samples collected along the Djerba transect (DJB-01 to -15) generally consist of Amphistegina lessonii, Gavelinopsis praegeri, Neoconorbina terquemi, Peneroplis spp. Planorbulina mediterranensis, varying in abundance from sample to sample. The number of species varies from 15 to 48 and Shannon Index ranges from 2.2 to 3.0.

Assemblage composition, documented in coastal stations, is very variable. Station CST-04 is barren of benthic foraminifera, whereas stations CST-06 and -07 contain the dominant Ammonia beccarii, and abundant A. parkinsoniana and Quinqueloculina laevigata. All other species (Supplemental Information S1) are rarer. The species number varies from 12 to 25 and the Shannon Index from 2.2 to 2.3. The more representative species identified in the Gulf of Gabes are documented in Figs. 6 to 11.

Figure 5 LINKTREE dendrogram showing the separation of GBS and DJB samples according to the major pollutants.

The SIMPROF Cluster analysis identified 6 clusters/groups within the Gabes and Djerba samples. All coastal stations including the foraminifera-barren coastal (CST-04) and Gabes sample (GBS-03) grouped separately (Figs. 2 and 3).

This outlier group was not used for additional statistical analyses because samples were either not analysed for the geochemical characterization (e.g., coastal stations) or were barren of benthic foraminifera (GBS-03 and CTS-04) or, in the case of all the CST, samples were collected in a different season (January: winter). Thus, assemblages that are linked to seasonality cannot be compared to those from the Gabes and Djerba transects collected in July (summer) 2014.

Figure 6 Images of selected foraminifera species.

(A–B) Ammoglobigerina globigeriniformis (Parker and Jones, 1865): (A) spiral view; (B) umbilical view. (C–D) Ammoscalaria runiuina (Heron-Allen and Earland, 1916): C spiral view; D umbilical view. (E–G) Clavulina difformis (Brady, 1884): E, G side view; F top view. (H–I) Eggerelloides scaber (Williamson, 1858): side view. (J–K) Glomospira charoides (Jones and Parker, 1860): side view. (L–M) Glomospira gordialis (Jones and Parker, 1860): L spiral view; M umbilical view. (N–O) Lagenammina fusiformis (Williamson, 1858): side view. (P–Q) Paratrochammina challengeri (Brönnimann and Whittaker, 1988): P spiral view; Q umbilical view. (R–S) Paratrochammina madeirae (Brönnimann, 1979): R spiral view; S umbilical view. (T–U) Trochammina inflata (Montagu, 1803): T spiral view; U umbilical view. (V–W) Psammosphaera fusca (Schulze, 1825): V spiral view; W umbilical view. (X–Y) Textularia sp.: side views. (Z–AA) Textularia conica (d’Orbigny, 1839): side views. BB-CC Textularia pseudorugosa (Lacroix, 1932). (DD–EE) Textularia agglutinans (d’Orbigny, 1839): side views. (FF–HH) Textularia calva (Lalicker, 1935): FF, HH side views; GG top view. (II–JJ) Mychostomina revertens (Rhumbler, 1906): II spiral view; JJ umbilical view.

Cluster A includes only sample GBS-06. This sample separates from Cluster B (samples GBS-07 to -08 and GBS-10 to -16) only because of the abundant presence of Textularia conica (66 specimens) that is otherwise rare (Table 3, Supplemental Information S1).

Cluster B is characterized by the highest contributions from E. scaber, B. elegantissima and B. striatula (Table 3). Bulimina marginata has a lower, yet similarly consistent contribution to the group.

Figure 7 Images of selected foraminifera species.

(A–B) Cornuspira involvens (Reuss, 1850): side view. (C–D) Patellina corrugata (Williamson, 1858): C spiral view; D umbilical view. (E–F) Vertebralina sp.: (E) spiral view; F umbilical view. (G–H) Wiesnerella auriculata (Egger, 1893): G spiral view; H umbilical view. (I–K) Articulina carinata (Wiesner, 1923): I, K side view; J top view. (L–N) Adelosina carinata-striata (Wiesner, 1923): L, M side view; N top view. (O–Q) Adelosina cliarensis (Heron-Allen and Earland, 1930): O, Q side view; P top view. (R–S) Adelosina laevigata (d’Orbigny, 1826) (juvenile):side views. (T–U) Affinetrina gualtieriana (d’Orbigny, 1839): side views. (V–X) Cycloforina contorta (d’Orbigny, 1846) (with a broken neck): V, W side view; X top view.; (Y–AA) Lachlanella variolata (d’Orbigny, 1826): Y, AA side view; Z top view. (BB–CC) Pseudotriculina laevigata (d’Orbigny, 1826): side view. (DD–EE) Quinqueloculina bidentata (d’Orbigny, 1839): side view. (FF–GG) Quinqueloculina bosciana (d’Orbigny, 1839): side view. (HH–JJ) Quinqueloculina jugosa (Cushman, 1944): HH, JJ side view; II top view.

Cluster C groups samples GBS-04 and -05 and has dominant contributions from Trochammina inflata and Rosalina villardeboana and lesser contributions by Q. laevigata, T. conica, B. striatula, B. elegantissima and Cornuspira involvens (Table 3).

Cluster D groups samples DJB-01 and -12 that are characterized by A. lessonii and to a lesser extent by R. villardeboana and P. mediterranensis (Table 3).

Figure 8 Images of selected foraminifera species.

(A–C) Quinqueloculina laevigata (d’ Orbigny, 1839): A, C side view; B top view. (D–F) Quinqueloculina pseudobuchiana (Luczowska, 1974): D, F side view; E top view. (G–H) Quinqueloculina seminula (Lineaeus, 1758): side views. (I–J) Miliolinella subrotunda (Montagu, 1803): side views. (K–M) Miliolinella webbiana (d’Orbigny, 1839): side views. (N–O) Biloculina (?) sp.: side views. (P–Q) Laevipeneroplis karreri (Wiesner, 1923): P umbilical view; Q spiral view. (R–T) Peneroplis planatus (Fichtel and Moll, 1798): R spiral view; S side view; T umbilical view. (U–W) Peneroplis pertusus (Forskål, 1775): U, W spiral view; V side view. (X–Y) Sorites orbiculus (Ehrenberg, 1839): X spiral view; Y umbilical view. (Z-AA) Carterina spiculotesta (Carter, 1877): Z spiral view; AA umbilical view. (BB–CC) Lagena striata (d’Orbigny, 1839) side view. (DD–FF) Lenticulina cultrata (de Montfort, 1808): DD, FF spiral view; EE side view. (GG–HH) Bolivina plicatella (Cushmann, 1930): side view. (II–JJ) Bolivina difformis (Williamson, 1858): side view. KK-LL Brizalina striatula (Cushman, 1922): side view.

Cluster E groups all the other samples from the Djerba transect (DJB-02 to -11, and DJB-13 to -15) that are predominantly characterized by Rosalina macropora and R. villardeboana and to a lesser extent by B. striatula, A. lessonii, A. mamilla and Peneroplis pertusus (Table 3).

Figure 9 Images of selected foraminifera species.

(A–B) Bulimininella elegantissima (d’Orbigny, 1839): side view. (C–D) Bulimina elongata (d’Orbigny, 1846): side view. (E–F) Bulimina marginata (d’Orbigny, 1826): side view. (G–H) Floresina sp.: side view. (I–J) Fursenkoina acuta (d’Orbigny, 1846): side view. (K–M) Siphogenerina raphana (Parker and Jones, 1865): K, L side view; M top view. (N–O) Sigmavirgulina tortuosa (Brady, 1881): side view. (P–R) Abditodentrix rhomboidalis (Millet, 1899): (P–Q) side view; R top view. (S–U) Uvigerina sp.: (S–T) side view; U top view. (V–W) Uvigerina canariensis (d’Orbigny): side view; (X–Y) Valvulineria minuta (Parker, 1954): X umbilical view; Y spiral view. (Z–AA) Glabratella altispira (Buzas, Smith and Beem, 1977): Z spiral view; AA umbilical view. (BB–CC) Facetocochlea pulchra (Cushman, 1933): BB spiral view; CC umbilical view. (DD–EE) Rosalina bradyi (Cushman, 1915): DD spiral view; EE umbilical view. (FF–GG) Rosalina globularis (d’Orbigny, 1826): FF spiral view; GG umbilical view. (HH–II) Rosalina macropora (Hofker, 1951): HH spiral view; II umbilical view. (JJ–KK) Rosalina pellucida (Said, 1949): JJ spiral view; KK umbilical view. (LL–MM) Rosalina villardeboana (d’Orbigny, 1839): LL spiral view; MM umbilical view. NN-PP Discorbinella concinna (Brady, 1884): specimen with floating chamber, NN spiral view; OO side view, PP umbilical view.

Cluster F groups samples GBS-01 to -02 and GBS-09. These are characterized by A. tepida and B. striatula (Table 3).

Discussion

Environmental conditions in the study area

The adverse environmental conditions in the Gulf of Gabes are well documented (e.g., El Zrelli et al., 2015; El Zrelli et al., 2017; El Kateb et al., 2016; El Kateb et al., 2018b and references therein). The research of El Zrelli et al. (2017) highlighted that the order of heavy metal concentrations in sediments follows Zn>Cd>Cr>Pb>Cu>Hg and that at least Zn and Cd derive from phosphogypsum. These authors observed that concentrations are higher close to the industrial complex and decrease toward the open sea. El Kateb (2018) and El Kateb et al. (2020) report high heavy metals (Zn, Cd, As, Cr, Fe, Cu) and phosphorus concentrations in the sea water as well as elevated Ptot and TOC in sediment samples collected close to the industrial complex at Gabes. The concentrations of the measured elements are higher along the Gabes transect than along the Djerba transect, with the exception of Zn. In particular, Zn concentrations in the sea water, along the Gabes transect (ranging between 7.05 ppm at GBS-11 and 64.63 ppm at GBS-16), are not as high as those reported by El Zrelli et al. (2017) in their coastal station sediments (ranging from 5.2 ppm along the coast to 7,165 ppm in front of the industrial complex). On the contrary it is relatively abundant in samples from the Djerba transect with a maximum of 104.75 ppm at DJB-15. El Kateb et al. (2020) propose that the low concentrations at Gabes could be related to the precipitation of Zn from the water into sphalerite (El Kateb et al., 2018b) and the subsequent transport of part of the remaining Zn off-shore during low tides (Othmani et al., 2017), while the high concentrations at Djerba could be related to the decay of plankton, which releases dissolved Zn into the seawater (Moore & Ramamoorthy, 1984).

Figure 10 Images of selected foraminifera species.

(A–B) Rosalina bulloides (d’Orbigny, 1839): A spiral view; B umbilical view. (C–D) Cassidulina obtusa (Williamson, 1858): C spiral view; D umbilical view. (E–F) Discorbinella berthelotti (d’Orbigny, 1839): E spiral view; F umbilical view. (G–H) Cibicidella variabilis (d’Orbigny, 1826): G spiral view; H umbilical view. (I–J) Gavelinopsis praegeri (Heron-Allen and Earland, 1913): I spiral view; J umbilical view. (K–L) Neoconorbina terquemi (Rzehak, 1888): K spiral view; L umbilical view. (M–N) Planorbulina mediterranensis (d’Orbigny, 1826): M spiral view; N umbilical view. (O–P) Planorbulina mediterranensis (d’Orbigny, 1826): O spiral view; P umbilical view. (Q–R) Cymbaloporetta bradyi (Cushman, 1915): Q spiral view; R umbilical view. (S–U) Asterigerinata mamilla (Williamson, 1858): S spiral view; T side view; U umbilical view. (V–X) Astrononion stelligerum (d’Orbigny, 1839): V, X spiral view; W side view. (Y–Z) Haynesina depressula (Walker and Jacob, 1798): Y spiral view; Z umbilical view. (AA–BB) Haynesina simplex (Cushman, 1933): AA spiral view; BB umbilical view. CC-DD Melonis affinis (Reuss, 1851): CC spiral view; DD umbilical view. (EE–FF) Nonion fabum (Fichtel and Moll, 1798): EE spiral view; FF umbilical view. GG-HH Nonionella turgida (Williamson, 1858): GG spiral view; HH umbilical view. (II–JJ) Noinionoides grateloupi (d’Orbigny, 1826): II spiral view; JJ umbilical view.

Based on the BIOENV and global BEST test, 5 of the measured environmental variables, cadmium (Cd), arsenic (As), phosphorus (P), total organic carbon (TOC) and total phosphorus (Ptot), were found to be best indices to separate (statistically significant at p <  0.001) the biotic (foraminiferal) assemblage data (ρ = 0.764). The LINKTREE plot (Fig. 5) shows that all the GBS samples cluster out from the DJB samples. The former all have elevated values in both the measured water (Cd, As, P) and sediment (TOC and Ptot) parameters.

Figure 11 Images of selected foraminifera species.

(A–B) Ammonia beccarii (Linnaeus, 1758): A spiral view; B umbilical view. (C–D) Ammonia convexa (Collins, 1958): C spiral view; D umbilical view. (E–F) Ammonia parkinsoniana (d’Orbigny, 1839): E spiral view; F umbilical view. (G–H) Ammonia tepida (Cushman, 1926): G spiral view; H umbilical view. (I–J) Amphistegina lessonii (d’Orbigny, 1826): I spiral view; J umbilical view. (K–L) Amphistegina lobifera (Larsen, 1976): K umbilical view; L spiral view. M-N Elphidium crispum (Linnaeus, 1758): spiral view. (O–P) Elphidium depressulum (Cushman, 1933): spiral view. (Q–R) Elphidium incertum (Williamson, 1858): spiral view.

Living foraminifera assemblages and their significance

Cluster A and B: polluted environment

These two clusters can be considered together as the only difference between the two is the abundant T. conica in Cluster A (which is composed by only one sample, GBS-06) and a different substratum (Fig. 3), Facies G3 for GBS-06, and Facies G4 for all stations of Cluster B. These two clusters group the majority of the stations in the Gulf of Gabes (Figs. 2 and 3).

The dominant foraminiferal species is E. scaber (Table 3). Murray (2006) described this species as infaunal, detritivore and typical of shelf environments. Bouchet (2007) suggests that the sedimentary detrital organic matter represents the main food resource for this species. Dessandier et al. (2016) studied the distribution of E. scaber along the Portuguese coast, highlighting its preference for environments characterized by high-quality of the organic matter (e.g., high chlorophyll-a to phaeopigment ratio and available amino acids). Murray (2013) reported that E. scaber is sensitive to salinity not exceeding 24 psu and oxygen depletion (<0.5 ml/L), and that it is tolerant to fluctuations in temperature and heavy metal pollution. Other highly contributing species of Cluster A and B are B. elegantissima and B. striatula, two stress-tolerant species able to survive in polluted environmental conditions and under oxygen depletion (e.g., Murray, 2006; Dimiza et al., 2019, and references therein). Martins et al. (2016a) documented the preference of B. striatula for high quality organic matter enriched in proteins, carbohydrates and chlorophyll -a.

In general, taxa which are tolerant to such conditions are typically associated with muddy substrates that can accumulate high amounts of organic matter (e.g., Van der Zwaan et al., 1999 and reference therein). This is evident as Cluster B includes stations from Facies G4 (clay with centimetric biogenic fragments). For Cluster B, the SIMPER analysis additionally identified the stress-tolerant species A. tepida and B. marginata (e.g., Dimiza et al., 2019; Jorissen et al., 2018), as well as Haynesina depressula, and Nonionoides grateloupi, which are classified as “indifferent species” by Jorissen et al. (2018). The relatively high contribution of A. mamilla in Cluster B (Table 3), in addition to its high abundances in samples from Clusters A and B (Supplemental Information S1), is not consistent with its life strategy from the literature. Murray (2013) described this species as epifaunal clinging on large detrital fragments or on marine vegetation. Dimiza et al. (2016) show the negative correlation between abundances of A. mamilla and the percent of mud. Is the quality of the organic matter playing a role on the abundance of this species? The abundance of this species may be, indeed, linked to the high quality of the organic matter, whose δ13C signature corresponds to marine phytoplankton, along the Gabes transect (Supplemental Information S2; El Kateb et al., 2018b; El Kateb et al., 2020). Alternatively, the centimetric bioclasts characterizing Facies G4 may provide a suitable ecological niche for A. mamilla (Fig. 3, Table 2).

The LINKTREE (Fig. 5) shows that stations grouped in Cluster A and B separate based on the relatively high (in relation the Djerba samples) values of TOC (≤3.34 wt. %), As (≤1.26 µg/L), Cd (0.73-1.19 µg/L) and phosphorus both in the sediments (87.1 > Ptot > 28.7 µmol/g ) and sea water (P > 144 µg/L).

In conclusion, the living (stained) benthic foraminifera assemblages from Clusters A and B are interpreted to reflect high sedimentary organic matter supply, in a stressed environment due to heavy metals and phosphorus contamination deriving from the Gabes industry.

Cluster C: polluted Environment characterized by physicochemical variability

Major contributors of this cluster (exceeding 13%) are T. inflata and R. vilardeboana (Table 3). It groups only two stations (GBS-04 and -05) which are characterized by sandy carbonate sediments and large biogenic fragments (Facies G3 in Table 2, El Kateb et al., 2018b) that may represent an ideal habitat for R. villardeboana. This species is described as stress-tolerant in culturing experiments (Hintz et al., 2004) and has been reported living in shallow water (Erginal et al., 2013), similarly to station GBS-04, with a water depth not exceeding 12 m (El Kateb et al., 2018b). Trochammina inflata is a species known to be highly tolerant to physicochemical variability (Murray, 2006; Martins et al., 2016b). It is a cosmopolitan epifaunal or infaunal, euryhaline marsh species, living down to 60 m of water depth and with a general preference for muddy substrates (Usera et al., 2002; Murray, 2006; Benito et al., 2016). The abundance of T. inflata within these samples may be due to the physicochemical variability linked to the tides that in this region reach the highest variations (up to 2 m, Aloulou, EllEuch & Kallel, 2012). Other species of this cluster are the stress-tolerant B. striatula, B. elegantissima and C. involvens, attributed to group 3 third-order opportunists, tolerant to and favoured by the first stages of organic enrichment (Alve et al., 2016; Jorissen et al., 2018). Rare specimens of Q. laevigata and T. conica (not exceeding 3 and 13 specimens, respectively) also occur, these species are considered as “sensitive” by Jorissen et al. (2018).

Cluster C is characterized by low species and low specimen numbers. In the LINKTREE (Fig. 5) it groups with Cluster A and B with relatively high TOC (≤3.34 wt. %), As (≤1.26 µg/L), Cd (0.73–1.19 µg/L) and high phosphorus concentrations both in the sediments (87.1 < Ptot >  8.7 µmol/g) and the sea water (P > 144 µg/L). In summary, the living (stained) benthic foraminifera assemblages from Cluster C represent a stressed environment under the impact of heavy metal and phosphorus contamination deriving from the Gabes industry. This is in combination with unstable physicochemical conditions that may account for the presence of more sensitive species.

Cluster D and E: Seagrass meadows and “pristine” sites

These clusters group all stations from the Djerba transect (Figs. 2 and 3). The living (stained) foraminifera assemblages of Clusters D and E show significant differences in comparison to the other clusters (Fig. 3, Table 3), highlighted by the consistent presence of sensitive species such as A. lessonii. Amphisteginids are large symbiont-bearing foraminifera widely used as bioindicators to assess water quality due to their fast response to environmental changes, e.g., water temperature and photo-oxidation (e.g., Emrich, Martínez-Colón & Alegria, 2017, De Freitas Prazeres, Martins & Bianchini, 2012; Hallock et al., 2006; Spezzaferri et al., 2018). Water transparency is an essential factor for Amphistegina spp. because these organisms host symbionts and, as such, are light dependent (Hallock, 1981). El Kateb et al. (2018c) suggest that the presence of P. oceanica meadows along the Djerba transect promotes the development of A. lessonii. This species is also very sensitive to heavy metals pollution because these pollutants directly affect its symbiotic algae. Furthermore, De Freitas Prazeres, Martins & Bianchini (2011) demonstrated, in culture, that zinc exposure causes visual alteration such as bleaching and/or dark brown areas in the test.

Cluster D is characterized by A. lessonii and to a lesser extent by R. villardeboana, P.  mediterranensis and R. macropora and includes two samples DJB-01 collected in the P. oceanica meadow (Facies D1) and DJB-12 (Facies D3, well-preserved biogenic fragments of bivalves, calcareous algae, gastropods and bryozoans), characterized by abundant degraded algae. R. vilardeboana is especially abundant at station DJB-01, where P. oceanica traps the sediments in its rhizomes to form small barriers. Planorbulina mediterranensis is epifaunal clinging on hard substrata and/or seagrass (Villanueva Guimerans & Cervera Currado, 1999). Mateu-Vicens et al. (2010) mentioned high abundances of this species in P. oceanica meadows, which is corroborated at station DJB-01 where the seagrass is densely covering the sea floor (El Kateb et al., 2018b). As an epifauna foraminifera, R. macropora is generally associated with sandy and vegetated substrate (Vidović et al., 2014 and references therein). Elshanawany et al. (2011) proposed R. macropora to be a species relatively sensitive to pollution.

Cluster E is characterized primarily by R. macropora, R. villardeboana in addition to the species B. striatula, A. lessonii, A. mamilla, P. pertusus. It includes all the remaining samples from Djerba covering a variety of Facies, from D1 to D3, ranging from P. oceanica meadows to fine sediment with biogenic clasts of various sizes. Similarly to the epibenthic and sensitive A. lessonii, P. pertusus is also a symbiont-bearing species and needs high light intensities (Samir & El-Din, 2001 and references therein). Amao, Kaminski & Setoyama (2016) suggested that this species is associated with the presence of marine flora. There are no investigations showing the pollution impact on this species but as a symbiont-bearing foraminifer, it can be considered as sensitive to anthropogenic-related stressors such as heavy metals. Within this cluster the occurrence of A. mamilla, R. vilardeboana and R. macropora seems to be linked to P. oceanica meadows, which provides them with an ideal substratum. Brizalina striatula is an infaunal species relatively tolerant to oxygen depletion and feeding on organic detritus (e.g., Murray, 2006). El Kateb et al. (2018a) documented abundant B. striatula linked to P. oceanica meadows, indicating that the remains of P. oceanica may represent a potential food resource for this species.

Table 3 Statistical analyses.

SIMPER analysis using the Bray–Curtis similarity matrix on fourth-root transformed living (stained) benthic foraminifera abundance data from the Lesser Syrtis. The main species contributing to the similarity within the SIMPROF cluster groups are identified. Note: Only the taxa contributing >5 to the within-group average similarity are shown.

Cluster	Species	Average Abundance	Average similarity	Similarity/SD	Contribution	
B	Eggerelloides scaber	2.96	6.84	9.93	11.95	
Buliminella elegantissima	2.43	5.65	9.28	9.87	
Brizalina striatula	2.27	5.22	9.29	9.12	
Asterigerinata mamilla	2.23	4.98	6.45	8.71	
Haynesina depressula	1.67	3.56	4.11	6.22	
Ammonia tepida	1.73	3.45	3.87	6.03	
Nonionides grateloupi	1.43	3.00	5.20	5.25	
Bulimina marginata	1.29	2.92	9.65	5.10	
C	Trochamina inflata	1.69	7.61	N/A	13.68	
Rosalina vilardeboana	2.04	7.27	N/A	13.07	
Quinqueloculina laevigata	1.25	5.78	N/A	10.40	
Textularia conica	1.54	5.78	N/A	10.40	
Brizalina striatula	1.48	4.86	N/A	8.74	
Buliminella elegantissima	1.28	4.86	N/A	8.74	
Cornuspira involvens	1.09	4.86	N/A	8.74	
D	Amphistegina lessonii	2.90	4.55	N/A	8.70	
Rosalina vilardeboana	4.08	4.36	N/A	8.33	
Planorbulina mediterranensis	2.43	3.75	N/A	7.17	
Rosalina macropora	2.21	3.07	N/A	5.86	
Mychostomina revertens	1.79	2.72	N/A	5.19	
E	Rosalina macropora	1.99	5.29	6.76	9.83	
Rosalina vilardeboana	1.57	4.11	4.57	7.65	
Brizalina striatula	1.55	4.01	4.76	7.45	
Amphistegina lessonii	1.77	4.00	1.87	7.44	
Asterigerinata mamilla	1.58	3.50	2.08	6.51	
Peneroplis pertusus	1.41	2.85	1.36	5.30	
F	Ammonia tepida	1.49	13.12	8.79	36.52	
Brizalina striatula	1.24	13.12	8.79	36.52	

All the Djerba sites are characterized by the highest species numbers, specimen numbers and Shannon diversities (Supplemental Information S1). They separate in the LINKTREE (Fig. 5) based on the lowest values of P, Ptot and Cd (<17.6 µg/L, <0.62 µmol/g and 23.5 µg/L, respectively) confirming the more negative association of the species, identified in the SIMPER analysis for Clusters D and E, with lower levels of pollution.

In summary, benthic foraminifera of Cluster D and E seem to be linked to P. oceanica meadows. Sensitive species such as A. lessonii and peneroplidae suggest relatively “pristine” environments where anthropogenic stress is minor (Triantaphyllou et al., 2005; Koukousioura et al., 2012).

Cluster F: Major industrial impact

This cluster groups Sample GBS-01, −02 and −09. The first two samples are located in front of the industrial complex at Gabes and are the closest to the pollution source, whereas sample GBS-09 is located further away. Sample GBS-02 was collected in the industrial sludge consisting of dark siliciclastic sediments (Facies G1, Fig. 3, Table 3), where concentrations of TOC and heavy metals reach the highest values in both seawater and marine sediments (Supplemental Information S2; El Zrelli et al., 2015; El Zrelli et al., 2018; Ayadi, Aloulou & Bouzid, 2015; El Kateb et al., 2018b). Major contributors to this cluster are A. tepida and B. striatula, two stress-tolerant species. In particular, A. tepida is well known to tolerate pollution and elevated concentrations of TOC (e.g., Samir & El-Din, 2001; Armynot du Châtelet, Debenay & Soulard, 2004; Dimiza et al., 2019). This cluster is characterized by the lowest species numbers, specimen numbers and Shannon diversities (Supplemental Information S1). The grouping of Sample GBS-09 with -01 and -02 is possibly due to the fine nature of the sediments that favors the accumulation of organic matter resulting in high TOC concentrations (Facies G5, Table 3, Figs. 3 and 5) (e.g., Tyson, 1995; Bergamaschi et al., 1997). Samples GBS-02 and −01 are still distinguished by their strong relation to high concentrations of Ptot, As and Cd (>282 µmol/g, >2.9 µg/L and 2.61 µg/L, respectively).

The living benthic foraminifera of Cluster F can be interpreted as a stress-tolerant assemblage related to high pollution levels (P, As, Cd) in a eutrophic setting.

Validation of results

El Kateb et al. (2018b) documented, in the target region, different environmental conditions characterized by differing geochemical parameters such as TOC, phosphorous content and grain size. Nevertheless, some dominant foraminiferal species are present over a large part of the studied area and are common in the Mediterranean Sea, both in the open sea and lagoons (Martins et al., 2015; Martins et al., 2016a) and, therefore, it is difficult to assign a specific environmental preference to them (e.g., A. mamilla, B. striatula). However, the distribution and abundance of the most representative species, illustrated in plots (Fig. 4) clearly shows that the clusters are related to their ecological preferences. For example the stress-tolerant species E. scaber, B. elegantissima, B. striatula, B. marginata, A. tepida are primarily abundant in the same clusters (Cluster A, B, C and F), which correspond to the GBS samples. On the contrary, epibenthic species, such as A. lessonii, R. macropora, R. vilardeboana and peneroplids are also most abundant in Clusters D and E. The consistent species distributions and clustering of stations, according to their foraminiferal content and geochemical/facies characterization (Figs. 2 and 5), confirms that the interpretation of species ecological preferences is plausible. It also highlights that species distributions are linked both to pollution and substratum type (Basso & Spezzaferri, 2000).

Conclusion

The Gulf of Gabes, in the Lesser Syrtis (Tunisia), is presently characterized by intense pollution from the phosphate treatment industries. This quantitative investigation on benthic foraminiferal assemblages has revealed the presence of six clusters representing: polluted settings (Clusters A and B); polluted environments characterized by physicochemical variability (Cluster C), seagrass meadows and “pristine” sites (Cluster D and E) and the area characterized by the major industrial impact (Cluster F).

The major pollutants in this region are phosphorus, organic matter, and heavy metals, in particular As and Cd, affecting benthic foraminiferal assemblages at various degrees. TOC has also an important role in controlling the assemblage composition and species diversity of Cluster F, which is dominantly composed of stress-tolerant species. Eighteen species are dominant but only ten show high statistical contributions to the clusters. The most stress-tolerant species are A. tepida (Cluster F) accompanied by E. scaber, B. elegantissima, B. striatula (Cluster A and B). Trochamina inflata may be interpreted as an indicator of physiochemical variability in the environment (Cluster C). Amphistegina spp., peneroplidae, P. mediterranensis, R. macropora and R. villardeboana can be considered as sensitive species living in P. oceanica meadows or clinging on coarse substratum (Clusters D and E).

Supplemental Information

Supplemental Information S1 Quantitative raw data of benthic foraminiferal assemblages, species number and Shannon (Hlog′). Modified after El Kateb et al. (2020)

Click here for additional data file.

Supplemental Information S2 Geochemical data

Click here for additional data file.

Supplemental Information S3 Taxonomical list of benthic foraminifera identified in this study

Click here for additional data file.

We are especially grateful to Kamel El Kateb for the organization of the field assessment in the Gulf of Gabes and to Christoph Neururer for his help during the sampling.

Additional Information and Declarations

Competing Interests

Author Contributions

Field Study Permissions

Data Availabilty

The authors declare there are no competing interests.

Akram El Kateb, Stephanie Stainbank and Silvia Spezzaferri conceived and designed the experiments, performed the experiments, analyzed the data, prepared figures and/or tables, authored or reviewed drafts of the paper, and approved the final draft.

Valentina Beccari conceived and designed the experiments, performed the experiments, analyzed the data, prepared figures and/or tables, authored or reviewed drafts of the paper, formatting, interpretation, and approved the final draft.

Giovanni Coletti conceived and designed the experiments, performed the experiments, analyzed the data, prepared figures and/or tables, authored or reviewed drafts of the paper, formatting, and approved the final draft.

The following information was supplied relating to field study approvals (i.e., approving body and any reference numbers):

No permissions were required to collect samples in the working area. All the locations where the samples were collected were publicly accessible, not specially protected, and not environmentally sensitive. The field studies does not involve endangered or protected species.

The following information was supplied regarding data availability:

All raw data related to this research are available in the Supplementary Files.

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
