# Peer review of "Living (stained) foraminifera in the Lesser Syrtis (Tunisia): influence of pollution and substratum"

_PeerJ, doi:10.7717/peerj.8839_

## Round 0.1 · original submission · Major Revisions

Dear authors,

We kindly invite you to analyze in depth the peer reviewers' comments and observations, especially those which recommended manuscript rejection, and reconsider your work in a consistent way. Please send us the improved paper within the recommended time frame, accompanied by explanations addressed to peer reviewers on how you responded to their observations.

Thank you!

Reviewer 1 ·

Basic reporting

No Comment

Experimental design

No Comment.

Validity of the findings

No comments

Additional comments

The Manuscript ID 41510-v0, entitled “Living (stained) foraminifera in the Lesser Syrtis (Tunisia):
influence of pollution and substratum” of the authors Akram El Kateb, Valentina Beccari, Silvia Spezzaferri, and Giovanni Coletti, is an interesting work. It aims to evaluate the negative impact of phosphate treatment industries along the Gulf of Gabes (Lesser Syrtis, Tunisia) on the marine environment, based on living foraminifera.
The work is based on the analysis of a significant number of stations located in well selected areas depending on proximity to source area of pollutants. The authors apply an excellent methodology for acquisition of data of living benthic foraminifera and present the location of the study sites and the foraminifera database. The work contains new aspects.
The text is presented in a manner that scientists in other disciplines will understand. It is presented and arranged clearly and concisely.
In the methodology, the authors should mention how they obtained authorization to sample in the studied area and which authorities gave permission to sample in Tunisia.
Many species of foraminifera have been photographed and the photos are very good allowing to clearly identify the photographed specimens. The identification of foraminifera is presented as supplementary material which is therefore of great interest.
Figure 3 is not readable. It should be subdivided and the information contained therein readable. The authors apply statistical analyzes and identify groups of stations characterized by differentiated composition of living foraminifera assemblages.
The work does not present physicochemical data measured in water and sediment nor sedimentological, such as particle size, TOC, metal concentrations. These data are important for explaining foraminifer-based station groups. If possible, this information should be added. The lack of this information puts the work at the level of a foraminifera checklist, which by itself is important.
The authors based the interpretation of station groups on the basis of living foraminifera species ecology citing old works based on total association (living+dead) (eg Vilela et al.). That methodology is contested by the FOBIMO group. The works based on total association cited and used to explain the ecology of the species must be changed by others analyzing living assemblages.
Works with living foraminifera in the Bizert Lagoon should be cited e.g.:
Martins, M.V.A et al. 2016. Organic matter quantity and quality, metals availability and foraminifera assemblages as environmental proxy applied to the Bizerte Lagoon (Tunisia). Marine Pollution Bulletin 105, 161–179, https://doi.org/10.1016/j.marpolbul.2016.02.032.
Martins, M.V.A. et al., 2015. Environmental quality assessment of Bizerte Lagoon (Tunisia) using living foraminifera assemblages and a multiproxy approach. PLoS ONE. http://dx.doi.10.1371/journal.pone.0137250

I think that the provided comments should be considered.

Reviewer 2 ·

Basic reporting

Language needs to be polished in several parts of the paper. In several parts of the discussion, paragraphs need to be rearranged in a way that the environmental conditions are described first and then the micropaleontological data are presented in order to support the various environmental conditions.
Several references to be added would improve the documentation.
Fig. 3 needs to be enlarged.
For further comments, see the attached pdf.

Experimental design

The originality of the research is well justified and research question is well defined.
Geochemical data need to be included and a correlation between main geochemical components and species deistribution will better support the foraminiferal environmental preferences. For further comments, see the attached pdf.

Validity of the findings

No comment

Annotated reviews are not available for download in order to protect the identity of reviewers who chose to remain anonymous.

Reviewer 3 ·

Basic reporting

- Professional English language was used
- References are updated and relevant
- The structure of the manuscript is adequate; figures are relevant (plots of Figure 3 are hard to be read); Table 1 needs to be improved with water depth of sampling stations raw data are shared
- This study is not self-contained because only the use of abiotic data publushed elsewhere (El-Kateb et al. 2018) would support conclusions

Experimental design

- The research is original and falls in the aim of the journal
- The research question is not clear. The Authors declaired the aim to "to trace the pollution produced by the phosphate industries in the Gulf of Gabes, to document assemblage composition variations in relation to the pollution sources" but they did not supply quantitative data on sediment and water parameters.
- The investigation on Benthic Foraminifera was rigorously conducted, according to recent international protocol
- Details on statistical analysis are missing. In particular, methods about the preparation of dataset and statistical analysis should be supplied.

Validity of the findings

1. This study consists in a very detailed and accurate documentation of living Benthic Foraminifera (BF) in the marine coastal area of Lesser Syrtis. A valuable taxonomic work was done reporting synonymies; also the huge work of illustrating species in photographic plates is appreciable. The utility if this work is considerable because it supplies a baseline for the classification of foraminiferal species in the area.
2. The study was rigorously conducted as regards sampling methods and quantitative analysis of BF, according to the FOBIMO protocol.
However, taking into account that the aim of the study is “to use the response of living BF to trace pollution ….. to document assemblage composition variations in relation to pollution sources” I consider this study affected by many weaknesses described below.
1. Although water and sediment abiotic parameters were mentioned in the interpretation of clusters obtained on the basis of foraminiferal results, no quantitative data, were reported; these data are available for the same sampling stations and published by El-Kateb et al. (2018b). In my opinion, this approach is not suitable to reliably demonstrate the actual role of abiotic factors on BF, in order to meet the aim of the study. Moreover, the response of BF to contamination should be demonstrated by a comparison of foraminiferal data with sediment/water data by means of suitable statistical techniques. Differently, the nMDS used by the Authors to enforce the environmental interpretation seems applied exclusively on foraminiferal data and the interpretation was given on the basis of the ecological features of foraminiferal taxa.
2. Also the attempt to supply environmental interpretation of clusters based on the autoecology of dominant taxa in each clusters was sometimes weak, because not supported by literature concerning the same species found in this study, but only dealing with the same genus (see below for more precise indications).
3. Quantitative data of species abundance (absolute or relative) were completely missing in the text of the manuscript (available exclusively from supplementary material).
4. Results on foraminiferal density and diversity, which are of basic importance while using BF for environmental assessment, were not reported at all.
5. The ecological interpretation of stressed environment for cluster A seems weak, because it was based on the supposed pollution tolerant character of Ammonia beccarii, while this species is very common in unstressed shallow-water Mediterranean environment (see below for references). Also reference to abiotic environmental parameters which may be the origin of such stressed conditions (phosphorous contamination, salinity changes) was not supported by quantitative and, on the whole, it is not clear which is the main environmental driver for this cluster.
On the whole, this study, as it is now set up, is not able to scientifically demonstrate what it was stated in the conclusions (the distribution of some species is related to different pollution degree), nor to achieve the purpose declared in the introduction (to document assemblage composition variations in relation to the pollution sources).

Additional comments

I have two main suggestions for re-addressing the work, because it could potentially be an excellent application of BF as bioindicators for the rigorous methodological approach adopted, but it needs considerable improvement finalized to support the given conclusions.
1. The interpretation of data should be carried out by integrating the biotic (in this study) and abiotic data (published in El-kateb et al. 2018b) by means of a suitable statistical approach.
2. I would also recommend the Authors to apply the Foram-AMBI index, according to Jorissen et al. (2018), and to compare the obtained values with the concentration of contaminants, in order to enhance the value of BF as tools for the environmental assessment.

Some minor observations are listed below.

Abstract
Lines 16-17. Because the audience of the journal is not limited to foraminifera specialists, I think that more preliminary information on BF as applied as environmental indicators should be supplied.
Line 21. I would specify “sediment samples”.
Lines 26-29. Which is the criterion for considering these species as good bioindicators? Please, specify.
Line 27. Eggerelloides scabrous is reported in the taxonomic list as Eggerelloides scabrus. However, according to the Worms database, the accepted name is Eggerelloides scaber.

Introduction
Line 49. Please, substitute “radioactivity” with “radioactive isotopes”.
Line 59. Zaouali et al. 1993 is not reported in reference list.
Lines 61-62. This sentence is not clear to me.
Lines 89-93. I would specify that the FOBIMO protocol was aimed at the standardization of methods for the application of BF in biomonitoring activities for the assessment of the ecological status, according to the Marine Strategy Framework Directive (MSFD). The cited papers may not be considered as an assessment of the contamination impact on marine environments, but studies on the response of living BF to specific environmental factors; in particular, Polovodova et al. (2015) did not consider living BF because it studied samples from sediment cores.
Lines 95-95. What does it mean “to trace pollution”? Please, clarify.
Although several earlier works examined sediment contamination in the study area, no quantitative concentration data for specific contaminants were reported from literature. More detailed reference to these works, including their location with respect to the present one would be needed.

MATERIALS AND METHODS
Because seasonality largely influences the living assemblage, the sampling period should be specified (and should be the same for all samples).
Lines 108-112. The range of water depth of transect stations should be supplied.
Details on the preparation of dataset (were all species included or rare species were not considered? Which was the principle for defining a species as rare?) and methods of statistical analysis should be given here.

DISCUSSION
Lines 152-153. I would not base the interpretation of stressed environment on the dominance of Ammonia beccarii, because assemblages dominated by this species are common in Mediterranean shallow-water unpolluted environment (see for example Walton and Sloan 1990 Journal Foraminiferal Research 20, 128-156; Murray 1991 Ecology and Paleoecology of Benthic Foraminifera. Longman Scientific & Technical). I would also suggest to critically consider literature reporting information on the tolerant character of this species because, especially in past times, several Ammonia species (mostly A. tepida and A. parkinsoniana) were classified as beccarii.
Lines 155-156. Concentration ranges of phosphorous should be supplied.
Lines 157-159. Because, in general, different species of the same genus have different ecological requirements, the ecological interpretation given by the Authors should be based on information regarding the species found in their samples, particularly Quinqueloculina laevigata, not “some quinqueloculinids”.
Lines 160-161. Salinity values should be supplied.
Lines 167-168. Please, supply quantitative data. Foraminiferal density is influenced by several environmental parameters such as sediment texture, nutrients, oxygen. How can the authors attribute the low foraminiferal density to contamination when these parameters were not considered?
Line 174. Please, write “Sen Gupta” instead of “Gupta”.
Lines 184-186. Which heavy metals showed high concentrations? Please, supply quantitative data. El-Kateb et al. (2018b) did not analyze heavy metals, but only mineralogical sediment composition.
Lines 191-192. Are the Authors speaking of statistical correlation? Please, specify.
Line 202. Please, supply data on mud percentage.
Lines 204-206. Organic matter is a potential food resource for all foraminifera species. Which are conditions favoring the abundance of Asterigerinata mamilla?
Line 215-2016. I don’t understand how sediments samples may be considered as hard substrate. El-kateb 2018b reports results of grain size analysis for sample CST-05. I would not define this cluster as related to hard substratum, but to coarse bottom sediment.
Lines 251-252. Please, add a reference.
Line 256. Because the Authors carried out the Cluster Analysis, a two-way (Q and R-mode) plot would have been sufficient to identify the dominant species in each cluster, instead of a Non-metric MultiDimensional Scaling consisting of 15 plots very hard to be read.
Lines 267-270. How the authors compared abiotic data with the plot based on foraminiferal abundance? this is the crux of all the work, because this comparison must be conducted with a statistical approach and based on a matrix containing all the data. How the Authors obtained the scale of grain size and pollution at the base of the nMDS plot (Fig. 4)?
Line 275. Table 2 reports only geographic coordinates. Water depth should be included.
Lines 268-270. These statements are not supported by the results of this study. The role and importance of abiotic factors on BF may be demonstrated only applying a statistical analysis which considers together biotic and abiotic data. At least the range in each cluster of these parameters, published by El-kateb 2018b, should be supplied in this study.

---

## Round 0.2 · accepted · Accept

In the final proof you are invited to make the corrections recommended by the peer-reviewer, namely:

LINE 387. Field et al. (1982) was not reported in the reference list.
LINE 737. “Jorissen” instead of “Jorisse”.
CONCLUSION. I would say something concerning the important role of TOC on foraminiferal distribution

Reviewer 3 ·

Basic reporting

Clear and unambiguous, professional English was used throughout.
Literature references, sufficient field background/context were provided.
Professional article structure, figures (I’m still convinced that small writing of Figure 4 result unreadable), tables were supplied.
Raw data were shared.
I consider this second version of the manuscript self-contained with relevant results to hypotheses.
In my opinion, the study changed completely perspective with respect to the first version due to the introduction of abiotic water and sediment data. t is now coherent with the declared aim and, in the present version, the conclusions are adequately supported by the results. Also, the more careful discussion of the ecological significance of species improved the quality of the manuscript.

LINE 387. Field et al. (1982) was not reported in the reference list.
LINE 737. “Jorissen” instead of “Jorisse”.
CONCLUSION. I would say something concerning the important role of TOC on foraminiferal distribution

Experimental design

ok

Validity of the findings

The statistical analysis aimed at highlighting relationships between foraminiferal species and environmental parameters, carried out in the revised version of the manuscript, added scientific value to the study.

Additional comments

In my opinion, the study changed completely perspective with respect to the first version, due to the introduction of abiotic water and sediment data. The statistical analysis aimed at highlighting relationships between foraminiferal species and environmental parameters added scientific value to the study. It is now coherent with the declared aim and, in the present version, the conclusions are adequately supported by the results. Also, the more careful discussion of the ecological significance of species improved the quality of the manuscript. I appreciate very much the photographic plates.